# The Empirical Phenomenological Method: Theoretical Foundation and Research Applications

**Luigina Mortari \*, Federica Valbusa, Marco Ubbiali and Rosi Bombieri**

Department of Human Sciences, University of Verona, 37129 Verona, Italy; federica.valbusa@univr.it (F.V.); marco.ubbiali@univr.it (M.U.); rosi.bombieri@univr.it (R.B.)
**\*** Correspondence: luigina.mortari@univr.it

**Abstract:** Phenomenological philosophy was developed by Husserl for the eidetic sciences, which are interested in the general essences or persistent characteristics of things. By contrast, the empirical sciences are sciences of facts, interested in the concrete, singular, contextual and accidental qualities of phenomena. We do not encounter general, pure essences in concrete reality; instead, we meet phenomena, which present themselves as the particular actualisations of the essences. For this reason, it is legitimate to distinguish between the eidetic essence, which is constituted by a set of essential predicates that necessarily belong to the thing, and the essence of the concrete, which is constituted by a set of predicates that characterises that unique and singular thing in the space and time in which it manifests itself. Starting from these considerations, this article presents an original interpretation of Husserl's phenomenological method to develop an empirical phenomenological theory. The 'empirical phenomenological method' (EPM) grounded in this theory will first be described, and two examples of its application, in healthcare and educational research, will then be presented.

**Keywords:** empirical phenomenological method; qualitative research; healthcare; education; care

## 1. Introduction

Among qualitative methods, the phenomenological method occupies an important place (Barritt et al. 1983; Giorgi 1985; Van Manen 1990; Anderson 1991; Angus et al. 1991; Karlsson 1993; Moustakas 1994; Ray 1994; Creswell 1998; Cohen et al. 2000; Dahlberg 2006; Bengtsson 2013b; Zahavi and Martiny 2019). In the human and social sciences, no discipline has neglected phenomenology, including anthropology, pedagogy, psychology, sociology, the organisational sciences and the health sciences. The relevance of phenomenology can be explained by the following facts: (i) phenomenology takes, as its object of investigation, lived experiences, which are essential to comprehend human experience; (ii) phenomenology is guided by the effort to establish itself as a rigorous science, and this is an essential goal of research dealing with experience that is engaged in achieving clear, scientific reliability; and (iii) phenomenology is characterised as a philosophical method (Heidegger 2010, p. 26; Lévinas 1969, p. 28) and, since research that deals with human experience is in search of a rigorous method, phenomenology can be taken as a valid reference point. Nevertheless, the transposition of the principles of phenomenology, founded by Husserl as a philosophical method, to the empirical field requires further thought.

The establishment of phenomenology as a research philosophy in the empirical field occurred in the North American context as a consequence of the spread of the thought of Alfred Schütz (1899–1959). However, precisely in this context, phenomenology has often revealed, especially in the past, a lack of a direct connection to the original sources of phenomenological philosophy. Indeed, through a careful screening of the phenomenological literature, Cohen and Omery (1994) found that several researchers quote secondary sources, such as the representatives of the various schools ("Duquesne School", "Heideggerian Hermeneutics School" and "Dutch School"), instead of directly referring to the original texts by the founders of phenomenology.

The method of phenomenological philosophy was developed by Husserl (2012) for the eidetic sciences, which are interested in the general essences or persistent characteristics of things. By contrast, the empirical sciences are "sciences of 'fact'" (Husserl 2012, p. 10), interested in the concrete, singular, contextual and accidental qualities of phenomena. This raises the following questions: If the interest of empirical science differs from that of phenomenology, is it legitimate to apply the phenomenological method, which was founded as a philosophical method, to the empirical human sciences? Moreover, can the empirical sciences, although they are interested in situational qualities, integrate the research on essence to provide a rigorous basis for their research? And if they can, what kind of essence would it be? Is it possible to hypothesise two different concepts of essence?

To address these questions means to search for the form of application of the phenomenological method in the sciences of experience, and this is precisely the purpose of our contribution. Giving reference to the first period of Husserl's production, in particular, through the rediscovery and analysis of some key concepts, which are highlighted in the *Ideas* and until now have been little explored and deepened by the phenomenologists engaged in empirical research, we aim to present an original theory of empirical phenomenology. The empirical phenomenological method (EPM) (Mortari 2022) grounded in this theory will first be described, and examples of its application in healthcare and educational research will then be presented.

## 2. Phenomenology and Empirical Research

In recent years, a debate has arisen about how to interpret phenomenology in its original sense and develop an authentic phenomenological method for use in empirical research (Van Manen 2017b, 2017a; Zahavi 2019; Zahavi and Martiny 2019). Before briefly discussing some different positions, it is critical to point out that there are many approaches of phenomenology that have both commonalities and distinct features (Spiegelberg 1982; Dowling 2007). Many empirical research methods based on phenomenology have been developed, and they reference different phenomenologists and interpretations of their key concepts. Indeed, even if phenomenological philosophers did not develop empirical research methods, their epistemological theories have been often used to fortify qualitative research approaches (Fleming et al. 2003; Dowling 2007). In this paragraph, we briefly present some of the main approaches to phenomenological empirical research to highlight the main phenomenological concepts upon which they are based.

The main distinguishing feature of the so-called 'new' (Crotty 1996) or 'scientific' (Giorgi 2000) phenomenology, i.e., the phenomenology applied to empirical research, from the traditional philosophical phenomenology is to consider other people's experiences of phenomena instead of intending the phenomenological investigation as a solitary endeavour of the researcher. Giorgi (2000), for example, states: 'Since a situation does not exist "in itself" as a chair or a pen might, what is "shameful" or "good listening" [...] requires descriptions about situations from persons who experienced them in the manner they did' (p. 14). Giorgi (2000) does not agree with the idea that the purpose of scientific phenomenology is to study the 'subjective experience of the people' while the purpose of philosophical phenomenology is to study 'the *objects* of human experience' (Crotty 1996, p. 3). To avoid this error, Giorgi (2000) refers to the phenomenological idea of intentionality; in fact, the situations about which other people's descriptions are collected should be considered 'the intentional objects of a series of conscious acts on the part of the subjects' (p. 14).

Giorgi (2000) considers his descriptive phenomenological method, which is based on the work of Husserl and Merleau-Ponty, to be authentically phenomenological—but scientifically rather than philosophically. According to this method, participants' descriptions, which are often obtained through an interview, should be analysed as follows: (1) read for a sense of the whole, (2) establish meaning units within the texts, (3) transform the established meaning units into psychologically sensitive expressions, (4) practice imaginative variation on the transformed meaning units 'to see what is truly essential about them', and then

carefully describe 'the most invariant connected meanings belonging to the experience, and that is the general structure' (Giorgi and Giorgi 2003, pp. 251–53).

Karlsson developed an empirical phenomenological psychological method (Karlsson 1993) based on the Husserlian notion of the intentionality of consciousness. According to this method, the phenomenological researcher aims to describe the 'meaning structure' of the consciousness of a specific phenomenon, i.e., the characteristic traits forming people's understanding of it. These traits may consist of prejudices, attitudes, notions, thoughts and feelings that give form to the way in which people experience a phenomenon. Karlsson (1996) emphasises that a phenomenological analysis is not a mere phenomenal description because it aims to describe 'the logos of the phenomenon, that is to say those necessary constituents (structure) which are needed in order for just that particular phenomenon to be what it is' (p. 307). The researcher's role is to collect as concrete a set of descriptions as possible, reflect upon those descriptions and carry out an interpretative analysis (Hellström et al. 1999; Karlsson and Sjöberg 2009; Leiviskä et al. 2011). When analysing a text, the researchers must use epoché, i.e., remain attentive towards their tendency to interpret phenomena through familiar meanings, and identify both the typological structures, i.e., the different constellations of meaning that can be found in the data, and the general meaning structure of a phenomenon, i.e., the set of constituents common to all the collected data.

Moustakas (1990), who developed a phenomenological approach known as 'heuristic research', refers to Husserl's concepts of intentionality of consciousness, epoché, phenomenological reduction and imaginative variation. In this approach, researchers must be attentive to participants' experiences of phenomena while also reflecting on their own experiences of the phenomena. In fact, meaning is grasped when the object, as it appears in our consciousness, mingles with the object in the world: 'What appears in consciousness is an absolute reality while what appears to the world is a product of learning' (Moustakas 1994, p. 27). According to Moustakas (1994, p. 59), the phenomenological method is characterised by an autobiographic value because the researcher has to feel the investigated phenomenon as involving him/herself. However, the researcher is also required to be disciplined, and this equilibrium between personal involvement and discipline is reached through continuous reflection, which takes the form of a 'self-dialogue'. As for data-gathering instruments, Moustakas refers to not only descriptions but also narrations, journals, personal documents and artistic products that help participants express their personal experience. A specific aspect of Moustakas's approach is that it does not require a general structural description but a creative synthesis of the different singular descriptions collected from participants (Douglass and Moustakas 1985).

Van Manen (1990) is known for developing a hermeneutic phenomenological approach to human science research. According to Van Manen (1984), phenomenology is the study of the lived experience or the 'lifeworld'—that is, 'the world as we immediately experience it rather than as we conceptualise, categorise, or theorize about it' (p. 37). Typical phenomenological research questions are formulated as follows: 'What is this lived experience like?', 'What is it like to experience this phenomenon or event?' or 'How do we understand or become aware of the primal meaning(s) of this experience?' (Van Manen 2017a, p. 776). The purpose of a phenomenological inquiry is to reach 'meaningful insights', which are gained through the engagement in reduction 'practiced as a constant questioning', and which are characterised as 'inceptual' rather than conceptual (Van Manen 2017b, p. 819). Van Manen presents the notion of 'inceptuality', referencing Heidegger (Heidegger 1999, p. 45; Heidegger 2012, pp. 70–71), and clarifies that it may come to the researcher 'as a gift, a grace'—or in other words, an event that he could 'neither plan nor foresee' (Van Manen 2017b, p. 823). According to Van Manen (1984), using personal experience is the starting point of a phenomenological investigation; however, other kinds of material can be considered, including other people's experiences, biographies or reconstructed life stories, as well as artistic and literary sources containing experiential descriptions. Analysing such material requires thematic analysis, which consists of determining the

'experiential structures that make up' the investigated experience (Van Manen 1984, p. 59). The researcher is required to produce a descriptive text where a phenomenon is described 'through the art of writing and rewriting' (Van Manen 1990, p. 30), which is crucial because 'the more profound phenomenological insights may only come' by engaging in these activities (Van Manen 2017b, p. 823).

Dahlberg, who developed the reflective lifeworld research approach (Dahlberg et al. 2001, 2007; Dahlberg and Dahlberg 2003, 2004; Dahlberg 2006), refers to Husserl's claim that the phenomenon has an essential meaning, and that 'if the essential meaning changes in a certain way, it is a different phenomenon' (Dahlberg 2006, p. 13). However, she observes how, following Husserl, one might think that essences are opposed to particularities and, therefore, integrates his view with that of Merleau-Ponty, who underlines how the meaning of a phenomenon is revealed to us in its totality and in its relationship with its particulars (Dahlberg 2006, p. 13). Dahlberg conceives the research report as a description of the essence of the phenomenon and its constituents, which she connects to the Husserlian concept of 'individualisations of essence'. Her intent is to arrive at a complete description that includes both the essential structure and the constituents of the phenomenon, thus keeping all aspects together, from the most abstract to the most concrete ones. When using a phenomenological approach, the researcher must endeavour to search for meaning while maintaining an 'open and bridled' attitude (Dahlberg 2006; Dahlberg and Dahlberg 2019): Dahlberg chooses the term 'bridling' rather than 'reduction' arguing that it 'covers an understanding that not only takes care of the particular pre-understanding, but the understanding as a whole', and she specifies that this attitude is characterised by 'actively waiting' for the phenomenon (Dahlberg 2006, p. 16).

Bengtsson's (1984) approach considers the concept of the lifeworld, one of the most fruitful phenomenological concepts, and notes its presence not only in the latest of Husserl's productions but also in an early manuscript dated 1916–17. He also refers to Merleau-Ponty (1945) when arguing that the subject experiences the lifeworld as an embodied being (Bengtsson 2013a), an assumption that is especially appropriate in pedagogical practice because the subject is characterised as an agent rather than merely a spectator, as in Husserl's transcendental phenomenology. He also refers to Schutz (1962), who asserts that it is in face-to-face situations that the possibility of understanding the other's lifeworld is to be found. According to Bengtsson (2013b), phenomenological concepts 'must be adapted to the particular research question', and 'their purpose is to enable the researcher to identify and understand phenomena in a lifeworld sensitive way' (p. 8). Therefore, the 'lifeworld approach' does not present a rigid procedure but 'stimulates creativity of methods' (Bengtsson 2013b, p. 10). Bengtsson recognises that 'both the people who are studied and the researchers are inseparably embedded in their different lifeworlds' (Bengtsson 2013b, p. 9). Therefore, bridges 'must be built between the lifeworld of the researcher and the lifeworld of the participants of the study' (Bengtsson 2013b, p. 9).

Concerning the term 'phenomenology', Zahavi agrees with Van Manen in believing that an overly arbitrary use of it 'will lead to an erosion of the reputation of phenomenology' itself (Zahavi 2019, p. 900). However, Zahavi and Martiny (2019) strongly criticised 'hyperphilosophical', phenomenological research; in particular, they argue that epoché and reduction are fundamental, transcendental philosophical ideas but that they need not always be present in the application of phenomenology in non-philosophical fields (Zahavi and Martiny 2019; Zahavi 2021). Instead, other more relevant aspects should be considered, such as 'phenomenology's criticism of scientism and its recognition of the importance of the lifeworld', as well as 'its insistence on developing an open-minded and non-biased attitude' and 'its careful analysis of human existence, where the subject is understood as an embodied and socially and culturally embedded being-in-the-world' (Zahavi and Martiny 2019, p. 161). Basically, it is argued that it is necessary to shift from focusing on the orthodoxy of the method to its potential to produce certain results. In particular, various examples of applications of phenomenology that demonstrate its usefulness, can be appreciated, especially concerning interviews in the health field (Zahavi and Martiny 2019).

This overview, without presuming to be exhaustive, shows how phenomenology can fruitfully inspire empirical research approaches, which differ in their specificities even if they often share some common theoretical references. What is problematic is that most existing empirical phenomenological approaches have been drawn on concepts thought up by Husserl for phenomenology as an eidetic research method. Instead, our proposal aims to found a theory of an empirical phenomenology and, consequently, a method of empirical research, which is drawn on two concepts specifically used by Husserl to describe the ways of knowledge of concrete experience. These concepts, which have not yet been adequately explored in the phenomenological empirical literature, are those of concrete essence and subsequent adumbrations.

## 3. The Theory of Empirical Phenomenology

According to Husserl (2012), phenomenology 'has to do with "consciousness", with all types of experience, with acts and their correlates' (p. 2). So, the focus of phenomenology, as 'a science of "phenomena"' (Husserl 2012, p. 1), is the lived experiences [*Erlebnisse*] of the consciousness. One of the criticisms of the Husserlian phenomenology is that it does not consider the bodily dimension of reality enough; thanks to the reflection of Merleau-Ponty (1945), it is possible to answer this critique by arguing that, in the consciousness, all that is essential, including the bodily experience, remains.

However, eidetic phenomenology does not consider lived experiences in their concreteness but explores them to grasp their pure essences; by contrast, empirical phenomenology needs to remain bound to the concretely lived data. Indeed, a science of experience has as its object the flow of mental processes of consciousness '*in the concrete fullness and entirety with which they figure in their concrete context*' (Husserl 2012, p. 64).

Once it is established that the object is the same—i.e., the lived experiences of the mind—but considered in different ways, there are two further basic questions to be addressed: Is it possible to found empirical knowledge on the principle of searching for essences? If it is possible, what operation can be implemented to search for essences in the field of empirical research? To answer these questions means to take into account two concepts of phenomenological philosophy, i.e., those of eidetic essence and immediate intuition, which are problematic to apply in empirical research, and consequently to present the concepts of concrete essence and subsequent adumbrations, which are consistent with the application of the phenomenological method to empirical research.

### 3.1. The Eidetic Essence and the Concrete Essence

The eidetic, or invariant, essence of a thing is what is valid for everything to which the same nominal label can be attributed. However, since we do not encounter general, pure essences in the concrete reality but only phenomena that present themselves as the particular actualisations of essences, it is legitimate to distinguish between the eidetic essence, which is constituted by a set of essential predicates that necessarily belong to the thing, and the material essence or essence of the concrete, which is constituted by a set of predicates that characterise that unique and singular thing in the space and time in which it manifests itself. In this regard, Husserl (2012) states, 'On the one side stand the material, which in a certain sense are the *essences "properly so-called"*. But on the other side stands what is still eidetic but none the less fundamentally and essentially different: a *mere essential form*, which is indeed an essence, but a completely "*empty*" one, an essence which *in the fashion of an empty form fits all possible essences*' (p. 22).

As opposed to eidetic phenomenology, which searches for the eidetic essence—the general predicates—of a lived experience, the science of experience searches for its contingent qualities—the essence of the concrete—to build knowledge that embraces as many forms as possible of the concrete differentiations of the real. To collect many lived experiences, describe them and capture their concrete essence is the first step, but it is not sufficient to build science because to remain lost in the extreme differentiation of the real is not yet science. Therefore, the methodological proposal presented in this contribution hypothesises that a

science of experience is built by first acquiring knowledge of singular concrete essences and then providing a formulation of the extended essence of the concrete. The extended concrete essence is built by starting from the analysis of singular, concrete essences. It summarises the qualities common to the different elements of experience identified during an investigation.

### *3.2. Immediate Intuition and Subsequent Adumbrations*

Eidetic science is grounded on eidetic intuition, a cognitive act that understands invariant essences by immediately grasping them. In this process of apprehending the pure essence of a lived experience, there is neither analysis nor a slow construction of knowledge; instead, the mind is alone in front of the object and immediately receives its essential shape. Conceived in this way, intuition is not suitable for empirical research, because in the world of experience, understanding a phenomenon requires observing it repeatedly to carefully analyse data about it and construct knowledge step by step. The cognitive act that guides the empirical phenomenological investigation—which has as its object the concrete, and not the eidetic, essence of a lived experience—is not immediate intuition but the continuous attention required to engage in the method of 'subsequent adumbrations'. In this regard, Husserl (2012) states that everything '*can* be given "*one-sidedly*", whilst in succession more "sides", though never "all sides", can be given' (p. 12). The empirical knowledge of a phenomenon must be constructed gradually, and the method of subsequent adumbrations is useful for this purpose because it requires us to 'turn around' the phenomenon. The action of 'turning around' was conceived as a fundamental epistemic action already in Platonic epistemology, where the soul lets herself be carried around the things to contemplate them (Plato 1997, Phaedrus, 247c).

Given that attention, implied by the method of subsequent adumbrations, is the essential cognitive act of empirical phenomenology, it is now necessary to present the epistemic principles needed to orient attention during the heuristic process:

-   The principle of evidence, which requires that the process of inquiry remains faithful to the qualities that appear, and only to what appears, and therefore imposes on the researcher the need to speak of a thing '*only within the limits in which it then presents itself*' (Husserl 2012, p. 43);
-   The principle of ulteriority, which requires us to search for the modes of the real that remain shadowed or veiled, since every being has a proper original way of transcending its appearance.

To put into action effective attention, which is able to explore the evident and hidden aspects of a phenomenon to understand it in a faithful manner, it is important to engage in the epistemic posture of *epoché* (Husserl 2012, p. 59), that is, in bracketing all pre-given theories, beliefs and assumptions about the investigated phenomenon—more precisely, all the contents of consciousness that could affect our knowledge of it. This does not mean aspiring to an empty mind, because it is not possible to void the mind from the cognitive contents that structure it; rather, *epoché* should be conceived as an epistemic imperative, an idea that, requiring intellectual discipline, has a regulative function without demanding to be completely realised.

## 4. The Empirical Phenomenological Method

The theory of empirical phenomenology presented above provides a foundation for the empirical phenomenological method (EPM), which can be applied to understand the lived experiences collected through empirical research in the human sciences.

First, it is important to point out that the distinction between eidetic sciences and empirical sciences does not imply a disconnect in the investigative process because research on general essences is necessary to orient research towards concretely experienced essences. Indeed, the first act of an inquiry is to identify its object, which implies a clear definition of its eidetic essence. This requires answering the question, 'What is the phenomenon we intend to deal with?' To answer this question means to search for an essential, formal

definition that expresses the invariant qualities that structure the general quid of the investigated phenomenon. After this preliminary eidetic investigation, it is possible to carry out the heuristic actions and follow the methodological principles of the EPM, which are presented in Table 1. While heuristic actions are typically phenomenological, namely directly derived from the application in the empirical domain of certain Husserlian concepts, methodological principles—as well as the use of the terms of 'labels' and 'categories' to identify the products of the analysis—are common to other empirical research approaches, such as content analysis (Neuendorf 2017), grounded theory (Glaser and Strauss 1967; Charmaz 2014) and thematic analysis (Clarke and Braun 2021).

**Table 1.** The heuristic actions and methodological principles of the EPM.

| Heuristic Actions and Methodological Principles of EPM | | |
|---|---|---|
| Heuristic actions | (a) Gain access to concrete singular data | Determine how to collect data about the phenomenon whose essence is searched for, i.e., choose the most adequate instruments for recording participants' lived experiences. |
| | (b) Collect a plurality of lived experiences | Involve a plurality of participants with experiences of the investigated phenomenon to collect different actualisations of it. |
| | (c) Define the concrete singular essence of each lived experience | Describe the specific concrete qualities of each collected lived experience by creating a descriptive label for each collected datum. Make a list of the descriptive labels expressing the concrete singular essences of the collected lived experiences. |
| | (d) Build classes of similar data | Start from the list of the descriptive labels and cluster the similar ones (each class includes descriptive labels expressing similar concrete singular essences). |
| | (e) Formulate the first level of extended essences | Define with a concept (category label) the essence of each class of concrete singular essences, i.e., of each identified cluster of descriptive labels; this essence is defined as extended because it expresses qualities that belong to several collected lived experiences, i.e., to several variations of the investigated phenomenon. |
| | (f) Build a hierarchy of essences | Search for similarities between the first level of extended essences, build classes of them—i.e., cluster the previously elaborated category labels—and define with a concept the essence of each class (macrocategory label). Repeat this operation until achieving the most general level of essence that can be attained through an empirical process. In this way, a hierarchy of essences—or "a graded series of essences" (Husserl 2012, p. 25)—is built, which includes the most concretely dense to the most extensively abstract essences found during the analysis process. Elaborate the final coding system, which—by including the labels, categories and macrocategories formulated—puts into evidence the hierarchical relationships among the individuated essences, ranging from the absolutely individual and concrete to the most shared and abstract. |
| | (g) Recover and describe the absolutely unique data | Since not all data can be codified within a coding system, it is necessary to give value to the protruding (or outlying) data by commenting them in the final research report. |
| | (h) Elaborate the descriptive theory | Describe the qualities of the investigated phenomenon, starting with the most extensive to the most concrete essences, and then also considering the absolutely unique data. |
| | (i) Implement the principle of recursiveness | Continually return to the collected data and descriptive and conceptual labels formulated during heuristic actions c, e and f to verify the essential knowledge that is being built. |
| Methodological principles | First methodological principle ($\alpha$) | Identify quantitatively significant variations of the phenomenon. |
| | Second methodological principle ($\beta$) | Carry out a detailed analysis of each lived experience to bring to light its concrete singular essence. |
| | Third methodological principle ($\gamma$) | Reflectively supervise the cognitive processes underlying the heuristic actions. |

The product of a data analysis process carried out according to EPM is a "bunch" of essences, that can be traversed either in ascending order (from the most concrete to

the most extensive essences) or in descending order (from the most extensive to the most concrete essences). The "bunch" of essences, which is the fundamental outcome of the EPM, is graphically represented in Figure 1, which also shows that the eidetic essence, whose acquirement is the purpose of philosophical investigation, cannot be reached by empirical research and that some singular essences remain as unique data, as they cannot be categorized because of their particularities.

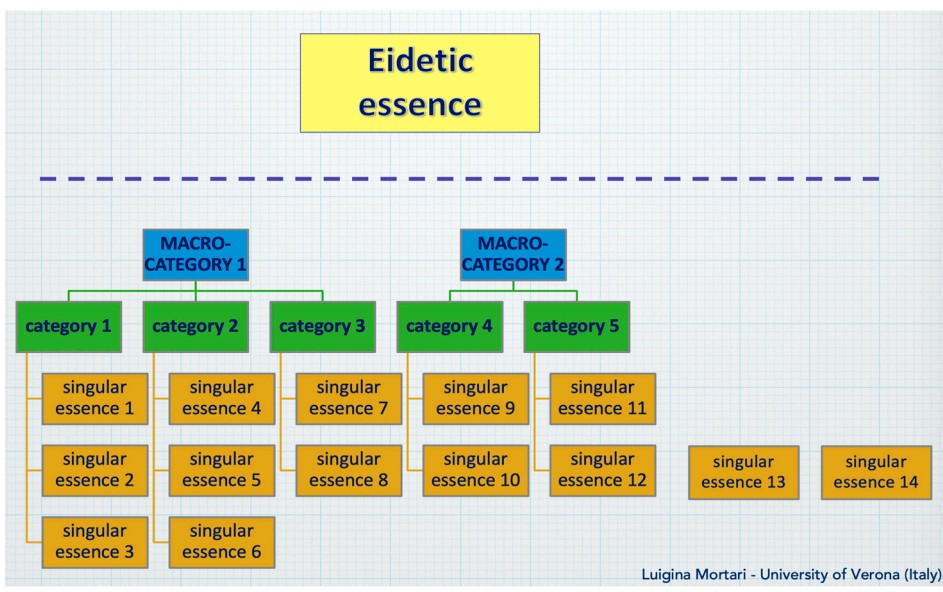

**Figure 1.** Graphic synthesis of the product of a data analysis process carried out according to the EPM.

In this inductively elaborated formulation of essences in a hierarchical structure, the difference between empirical phenomenology and eidetic phenomenology is evident. Indeed, Husserl, in defining the eidetic method, states that 'we cannot insist [...] on a systematic inductive procedure, on a gradual ascent, rung by rung, up the step-ladder of generality', and this is because 'it belongs to the very nature of a general apprehension of essences and of general analysis and description that there is no corresponding dependence of what is done at higher grades on what is done at the lower' (Husserl 2012, p. 144).

The application of the EPM has as its result a descriptive theory of the investigated phenomenon. This descriptive theory does not claim any absolute value because it is faithfully grounded in the collected lived experiences. This grounding in the investigated phenomena makes the descriptive theory seem to lack something, i.e., to fail to achieve that level of abstraction and generalisation that would guarantee it a scientific value. By contrast, it is precisely in this act of staying faithful to the datum that lies the evidence of the scientific rigor of the EPM.

## 5. Applications in Empirical Research

Following the theoretical foundation and explanation of the EPM, it is important to describe how it can be concretely applied in empirical research. For this purpose, we will present examples of its application in the fields of healthcare (Mortari and Saiani 2014) and educational research (Mortari and Ubbiali 2017; Mortari et al. 2017). These studies investigated the phenomenon of care as it is conceived by nurses, whose work consists in the everyday care of their patients, and by children involved in an educational project that required them to reflect on ethical concepts.

Since, as explained above, engagement in an empirical process requires a preliminary eidetic analysis of the phenomenon that the researcher intends to examine, the investigation was carried out on two levels: that of a theoretical inquiry, which for both of the studies took the form of a conceptual analysis of care, and that of qualitative empirical research, which in the first study considered concrete applications of care as they were described

by nurses working in healthcare structures and, in the second case, considered the ideas of care as they were expressed, in the form of descriptions and narrations, by children in kindergarten. The first level of the phenomenological investigation is the level of 'deskwork', which searches for the eidetic essence of the investigated phenomenon by taking into account its general and invariant qualities, i.e., the qualities shared by all its concrete actualisations; the second is that of 'fieldwork', which searches for the concrete essence of the investigated phenomenon by collecting individual lived experiences of it and analysing their specific qualities. Explaining how the EPM was applied in the two studies taken into consideration means focusing on the second level of investigation, i.e., on the empirical qualitative research, by giving a voice to nurses in the healthcare field and to children in the educational field.

### 5.1. The EPM in Healthcare Research

The first application of the EPM we would like to examine concerns qualitative research in the healthcare field, whose purpose was to identify what good care in nursing consists of. Below, the heuristic actions explained above will be exemplified with reference to the research method followed in this study.

a.     *Gain access to concrete singular data*

At first, we chose to use eidetic interviews, asking participants the following question: 'According to you, *what is* a good way of caring for a patient?'. Nevertheless, the answers collected turned out to be poor, being characterised by simplistic language, overly brief sentences and banal words; the participants used the knowledge learned in their formal training, only rarely reflecting on their own experiences. For this reason, we decided to change the research instrument, using narrative interviews and, accordingly, asking the participants new questions: 'Can you *narrate* an event, that occurred in your experience, in which you did a good action of care?' and 'Can you *narrate* an event, that occurred in your experience, in which you did a bad action of care?'. Even if phenomenology is a descriptive science and, in its original and traditional form, the phenomenological method required a collection of descriptions, collecting narrations should be considered epistemically legitimate because the essential quality of experience is temporality, as experience develops over time, and the cognitive form that seems to be most consistent with this characteristic of experience is narration. This choice is grounded in the literature: Bruner (1990) writes about a 'narrative truth', Muller (1999) highlights that narration contributes to creating meaning and Bateson (1979) suggests that all people think in terms of stories; also, phenomenological authors underline the importance of collecting data through narrations (Van Manen 1984; Moustakas 1990; Dahlberg 2006; Bengtsson 2013b). Moreover, research experience suggests that people involved in qualitative research generally find the task of narrating simpler than the task of describing, and a rich narration generally also includes detailed descriptions. Asking nurses for both positive and negative experiences of care, allows researchers to collect contrasting information, which enables them to construct clearer theories of the phenomenon of care in healthcare.

b.     *Collect a plurality of lived experiences*

The number of participants was not decided before the beginning of the research; instead, according to the principle of naturalistic inquiry, which recommends purposeful sampling (Merriam 2002), we decided 'in the field' when the number of participants with experiences of the investigated phenomenon was considered adequate based on the level of data saturation. The choice of the involved participants was made among people who had 'practical understanding' of the phenomenon, as recommended by Bengtsson (2013b). In the end, 120 male and female nurses working in different clinical areas in hospitals in Northern Italy were recruited. According to the research team's evaluation, this sample responded to the methodological principle ($\alpha$) of identifying quantitatively significant variations of the phenomenon.

c.    *Define the concrete singular essence of each lived experience*

The stories of good and bad caring experiences were recorded, transcribed verbatim and carefully read by the researchers. In carrying out the analysis, the research team worked in pairs, which were changed over time to allow a continuous exchange of perspectives. At the beginning, each pair worked autonomously to identify meaningful units within the text and label them, and then the two researchers worked together to compare labels until they agreed on a consensus version of them. This procedure responded to the methodological principle of carrying out a detailed analysis of each lived experience of nursing care to bring to light its concrete singular essence (β). An example of this heuristic action is presented in Table 2, where the meaningful units individuated by the researchers are highlighted in italics.

**Table 2.** An example of heuristic action c in healthcare research.

| Meaningful Unit | Descriptive Label (Researcher 1) | Descriptive Label (Researcher 2) | Descriptive Label Agreed on |
|---|---|---|---|
| Rose came to the delivery room at two in the morning, not worried about disturbing anyone. With her German accent, which made her demands sound even more peremptory, she asked to be helped to give birth in the way she wanted. *I think to myself how every case is different and, for this reason, hard to understand.* […] | She questions/reflects on her actions: the sign of a 'thinking presence'. | | She questions/reflects on her actions: the sign of a 'thinking presence'. |
| *I took her to the delivery room I thought most suitable for her.* […] | | She tries to find the most suitable environment. | She tries to find the most suitable environment. |
| *I wanted to isolate her as much as possible, or rather, I wanted to isolate myself as much as possible so as to assist her with the least possible interference. Rose wanted to be the absolute protagonist of the birth and did not want to be disturbed.* She described the welcoming ritual she wanted for her baby: [ … ] *She would wrap him in a red cloth (so the baby wouldn't notice the difference in the color of the new environment), and she would be the only one to touch him.* | Accepting a paradox, acting 'outside the box/norms'. She is receptive to the requests of the patient. She questions/reflects on her action: the sign of a 'thinking presence'. | | Accepting a paradox, acting 'outside the box/norms'. |

d.    *Build classes of similar data and*
e.    *Formulate the first level of the extended essences*

Starting from a list of descriptive labels, which express singular concrete essences of different actualisations of the phenomenon of nursing care, classes of these actualisations were elaborated. Each class, or category, refers to the concrete extended essence, which is shared by the singular lived experiences of nursing care grouped within it. Each extended essence is expressed by a concept or category label. While descriptive labels name the content of a meaningful unit of text, category labels name a cluster of descriptive labels grouped together based on analogy. An example of these actions is presented in Table 3.

**Table 3.** An example of heuristic actions d and e in healthcare research.

| Descriptive Label (Singular Concrete Essence) | Category Label (First-Level Extended Essence) |
| --- | --- |
| Keeping an eye on the patient | Paying attention |
| Listening | |
| Taking time to be with the patient | Dedicating time |
| Taking time to offer a caring word | |
| Being there in silence | |
| Using time for unforeseen actions | |
| Being capable of empathy | Understanding the other person |
| Interpreting the patient's experience | |

An effective way to put into action the third methodological principle (γ), which requires reflective supervision of the cognitive processes underlying the heuristic actions, is to keep a research diary in which the researcher's "life of the mind" (Arendt 1978) finds space to be described. Thanks to the reflective introspective writing, the researcher can give form to an inner dialogue, with a self-critical value (Lukinsky 1990; Taylor 1998). Moreover, in periodic supervision meetings with the team, not only the findings but also the pages of the diary dedicated to the most critical moments of the research process, became the object of a shared reflection (Johns and Hardy 1998; Knights 1985).

Below, we present an example of a diary entry that describes the cognitive supervision of heuristic actions c, d and e.

> One way to order data is to categorise them, to elaborate concepts within which we can arrange data. Elaborating concepts is an essential epistemic action. This, however, carries the risk that the concept neutralises the other's singularity. Conceptualising can exert a form of violence on the data, which, to be inserted in the concept, must abandon part of their singular givenness.

> When I formulate descriptive and then conceptual labels, can I find words that do not strip the other's saying of its otherness? Can I encounter their saying outside any a priori, any pre-structured hermeneutic asset? When I identify my excerpts, am I dismembering texts, grinding them inside pre-available hermeneutic grills? Or perhaps, by assigning labels to the quid of the text, do I manage not to lack respect for it?

f.    *Build a hierarchy of essences*

While descriptive labels are strictly grounded in data, the conceptual ones, category labels, represent a level of progressive abstraction that increases when we search for the second-level extended essences (macrocategories), i.e., when classes of the first-level extended essences are elaborated. The coding system presented below, which includes the elaborated descriptive, category and macrocategory labels, offers a synthetic description of what care is conceived to be by the nurses involved in the research, with an example of data for each label listed. The presentation of sample data is designed to highlight the adherence of the coding process to the concrete experiences recorded (Table 4).

**Table 4.** Coding system for the results of heuristic action f in healthcare research.

| Example of Data | Descriptive Label (Singular Concrete Essence) | Category Label (First-Level Extended Essence) | Macrocategory Label (Second-Level Extended Essence) |
|---|---|---|---|
| 'While carrying out my duties, I would now and then stop and look at her. Her look was always lost in space; she was curled up in bed under the sheets, as if she was in need of protection and tranquillity at such a difficult time . . . '. | Keeping an eye on the patient | Paying attention | Actions addressed to the patient |
| 'Antonietta had just been transferred from General Medicine to the coronary unit . . . I went to administer her treatment to her and felt she needed to talk. I had left her till the end of my 10 p.m. rounds so that I could sit and listen to her more carefully and without being interrupted'. | Listening | | |
| 'It was a frenetic day, and I was afraid I wouldn't find a way of going to see her . . . but eventually, I made it. I found some time, entered the room, opened the windows and smiled. She smiled back'. | Taking time to be with the patient | Dedicating time | |
| 'I found her on the bed by her mother; she was crying. I stayed there chatting with her for a long time, holding her hand'. | Taking time to offer a caring word | | |
| 'Returning to the ward after a two-day break, I entered the room. She asked me to close the door and immediately burst into tears. She was desperate. I got closer, and she hugged me. I returned the hug and sat on the bed waiting for her to stop crying'. | Being there in silence | | |
| 'One morning, she told me how she hated being without her make-up and nail polish; she felt uncomfortable being so scruffy. That afternoon, I went to a perfume shop and got her red polish, her favourite. I got her some nail varnish remover, too, just in case. Unfortunately, I never had a chance to give it to her, since during the night she went into a coma . . . '. | Spending time on unforeseen actions | | |
| 'I feel her pain, this woman, this mother; she is trembling both in body and soul. I feel she'd like an answer that would relieve her anguish . . . '. | Being capable of empathy | Understanding the other person | |
| 'Monica is a tracheotomised patient [ . . . ] I feel challenged by her. What can I, as a nurse, do in circumstances that I cannot change? . . . I [tried to] think of an alternative that doesn't pose a risk for Monica and at the same time meets her needs'. | Interpreting the patient's experience | | |
| 'Jacopo had fallen asleep. I put him in his crib and sat on the bed by his mother, searching for physical contact. I rested my hand on hers and spoke to her, trying to meet her red, swollen eyes . . . '. | Building a relationship with the patient through physical gestures | Trying to establish a relationship with the patient | |
| 'One morning, I found her in tears. I sat on her bed, put down the drip I had in my hand and asked her to tell me what was troubling her . . . She started telling me her story'. | Building a relationship with the other through verbal gestures | | |
| 'There's a patient who needs to sleep with his pillow turned in a certain way. The other one has to be put behind his back, and since he wasn't self-sufficient, we had to arrange his pillow for him at night. [ . . . ] To understand the real needs of others, you have to stop and listen, and then try again and again until you see an improvement. Eventually, he told me, "Now I'm fine. I can sleep, thanks". | Being receptive to the patient's personal requests | Satisfying the patient's needs | |
| 'The patient had been subjected to palliative cures for two days. She wanted to wash her hair, but she was afraid of feeling pain, so I volunteered to help . . . and to my surprise, she accepted'. | Helping the patients care for their body | | |
| 'The patient, treated for neoformation of the pharynx, expressed the need to go back to her habits after the operation—to wear a chador. I asked the doctor whether it would be possible for her to wear it on the recent surgery wounds. The doctor agreed, provided the rules of hygiene were complied with, and the patient started wearing her chador a few days after the operation'. | Helping the patients maintain their lifestyle | | |

**Table 4.** *Cont.*

| Example of Data | Descriptive Label (Singular Concrete Essence) | Category Label (First-Level Extended Essence) | Macrocategory Label (Second-Level Extended Essence) |
|---|---|---|---|
| 'You realise the pillow is warm, and you turn it over for her; the tissue that's just been placed by her mouth is already wet with saliva, so you change it'. | Soothing pain | Satisfying the patient's needs | |
| 'While waiting for the cardiologists, I went to his room and calmed him down, explaining the procedure'. | Calming | | |
| 'I reassure her that all the nurses will be briefed on her case history and that, in the case of any doubt and/or misunderstandings, she can contact me at any time, even after being discharged'. | Reassuring | Being concerned with the emotional dimension | |
| 'I was present at other crucial moments in her rehabilitation, such as the first time she got her legs out of bed or took her first steps. I comforted and encouraged her'. | Encouraging | | |
| 'During my first home visit, I thought Rosario looked neglected, just like the first time we met . . . I tried to put him at his ease. I didn't want to force him to care for his appearance. I had to gain his trust first'. | Building confidence | | Actions addressed to the patient |
| ' . . . I asked the patient to undress. She was embarrassed and replied that she was shy, so I offered to accompany her to the bathroom . . . '. | Preserving the patient's dignity | | |
| 'Eventually, I tried to take off the medication delicately with cod liver oil—not a centimetre, not half a centimetre, but a millimetre at a time'. | Acting with delicacy | | |
| 'When it was time to administer the infusion therapy to that patient, before entering his room, I would stand at the door for a fraction of a second, and that hiatus, however small, allowed his relatives to make room for me—an emotional space enabling the presence of another person to be accepted'. | Being present in a non-intrusive way | Respecting the other person | |
| 'So, on a number of occasions [ . . . ] whenever I felt the timing was right (when Gabriella was alone or even when her daughter was there . . . .), I started talking a little about this stoma'. | Adapting to the patient's pace | | |
| 'We decide to call the son of an intensive care patient whose condition was worsening. I let him sit down, and help him put on the gown and overshoes, and inform him that he's been called specifically on his father's request. He is anxious and worried and asks me to take him to his father's bedside. I ask him if he wishes to call the chaplain and whether he wants to accompany his father in the most difficult moment of his life: death. [ . . . ] I offer my support. Whatever his decision, I try to pose my questions in a discrete way. He stares at me and says, "Betty, I trust you, take me to him"'. | Impacting the relational context: improving the nurse–family relationship | Impacting the context to facilitate the act of caring | Attention to context |
| 'Despite the inflexibility of visiting times, we had decided that his wife and children could enter whenever possible, and sometimes we even allowed them to watch a film together'. | Impacting the organisational context | | |
| 'I expressed very clearly to the medical team what we thought about the way to manage a patient's pain, speaking passionately about the many moments we had spent by that patient's bed, and I asked for an anaesthesiology team to tackle the complexity of that pain'. | Building good relationships with colleagues and the medical team | | |
| 'I didn't know what to do. I seemed to have forgotten all my basic emergency and pharmacology knowledge learnt at school. [ . . . ] I prepared it with shaking hands, in the hurry to stop a pain that I felt was excruciating, but also because I was making an important decision, since from that moment F. would never be lucid and conscious again. [ . . . ] I heard my own voice talking to me, saying, "Come on, Chiara, go to him. Don't behave like this, nothing bad will happen"'. | Thinking about what to do | Thinking | Invisible caring |

**Table 4.** *Cont.*

| Example of Data | Descriptive Label (Singular Concrete Essence) | Category Label (First-Level Extended Essence) | Macrocategory Label (Second-Level Extended Essence) |
|---|---|---|---|
| 'I had never had this experience before, and I was a bit worried about how to deal with her. It turned out that she was my age. I knew her by sight, and I'd never have imagined I would see her like that. Reading her file, I saw what they wrote about her in A&E [Accident and Emergency Department] and I formed a rather harsh opinion, which I instantly tried to remove, concentrating on her need to be understood at that moment'. | Examining predetermined ideas | Thinking | |
| 'Immediately after the described event, I asked myself how it was it possible that a lung embolism hadn't been taken into consideration [ . . . ] I tried to work out what it was that had tipped me off, and it wasn't easy to find the answer'. | Questioning one's own actions | Reflecting on the experience | |
| 'His head was under the pillow, but I heard him sobbing . . . Then, I don't know what happened to me. I got scared. I didn't know how to approach him. I was afraid the patient would react negatively, that he would send me away or perhaps I was afraid he would ask me for help that I could not offer him. He would ask for hope . . . and I couldn't give him that, either. So, I didn't do anything, I literally ran away. I pretended not to see, as if he'd been sleeping'. | Assessing one's own actions | | Invisible caring |
| 'I keep looking at Roberto and tell myself that I shouldn't worry, that it is not professional to feel involved or moved by a patient. During my three years at university, I was taught to maintain detached empathy, which enables me to understand the patient's needs, but no more than that. The truth is that I keep looking at Roberto. and I feel afraid'. | Listening to one's own emotions | Dealing with one's own emotional experience | |
| 'What can I say now about the emptiness I felt at the news of her death? About the tears I shed on the day of her funeral? About how I hated myself and at the same time appreciated the fact of having strong feelings for a patient? I continuously wondered whether it was right for a nurse to get so close to a patient and asked myself, in my innermost thoughts, those that come and smother you suddenly at night, whether I had been a good nurse'. | Trying to handle one's own emotions | | |

The coding system highlights the hierarchy of essences identified through the analysis process; ranging from the most singular and concrete essences, defined by the descriptive labels, to the most extended and abstract ones, defined by the macrocategory labels.

g.   *Recover and describe the absolutely unique data*

In the case of this research, we codified the individuated meaningful units by elaborating descriptive labels for each one, which were then clustered into categories and macrocategories. Thus, no outlying data were found to be separately described and commented on in the research report as unique.

h.   *Elaborate the descriptive theory*

Finally, the application of the EPM in this research allowed the elaboration of a descriptive theory of the phenomenon of nursing care that emphasises the actions directed toward the patient, the actions carried out in the relational and physical contexts, and the invisible actions, i.e., the thinking and reflecting underlying the practice of caring. In the writing down of the theory, every macrocategory (i.e., second-level extended concrete essences), was presented with reference to the corresponding categories (i.e., first-level extended concrete essences) and descriptive labels (i.e., individual concrete essences), also highlighting the connection with the collected original data.

    i.    *Implement the principle of recursiveness*

During the heuristic process, the correspondence between the individuated meaningful units and the descriptive labels, category labels and macrocategory labels formulated was continuously checked with the aim of maintaining a theory of nursing care as faithful as possible to the collected data. This action, which guarantees the rigour of the qualitative research, benefits from the comparison of perspectives made possible by working in teams.

### 5.2. The EPM in Educational Research

The second example of applying the EPM concerns qualitative research in the educational field. Its purpose was to understand what care means according to kindergarten children. As with the study on care in the healthcare field, we exemplify the EPM heuristic actions below.

    a.    *Gain access to concrete singular data*

The instrument we chose to collect the data was the Socratic conversation, introduced by the following question: 'The word "'care'" is another beautiful word. What comes to your mind when you hear this word?' A Socratic conversation is known for being introduced by a question concerning the essential meaning of a phenomenon and conducted in a dialogical style inspired by the Socratic maieutic method: the researcher acts as a facilitator, stimulating children to express, clarify and deepen their thoughts. The question about care posed to children during this study was an open question formulated as such to make it possible to collect both eidetic, i.e., descriptive, and narrative data. Indeed, the children answered in two different ways: by directly describing what they meant by the word 'care' or by narrating experiences they conceived to be experiences of care, from which the researchers could indirectly infer what they meant by this concept. Often, the research diary kept by the researchers—an instrument that, as we explained above, effectively responds to the third methodological principle (γ)—included reflections about their ways of conducting the conversation in class, their dialogical posture and actions, and their ability to stimulate children's thinking without suggesting their own ideas about care.

    b.    *Collect a plurality of lived experiences*

Educational research can take the form of 'service research', i.e., research designed to respond to a concrete educative need individuated in an educational context. Consistent with this idea, this study was carried out in kindergartens that requested the involvement of the university to design and implement an intervention on ethical education. Qualitative research was then carried out to evaluate the intervention's effectiveness. The participants were 116 4–5-year-old children from kindergartens situated in North and Central Italy. According to the research team, this number of children responded to the first methodological principle (α), which requires obtaining a quantitatively significant variation of the phenomenon, which, in this case, was the children's idea of care.

    c.    *Define the concrete singular essence of each lived experience*

The conversational exchanges were audio-recorded, faithfully transcribed and carefully read by the researchers. The initial process of data analysis, responding to the second methodological principle (β) and consisting in individuating the singular essences of care expressed by the children's answers and describing them with synthetic descriptive labels, was first carried out by the researchers in pairs. The ambiguous cases were then discussed in teams. An example of this descriptive labeling action is presented in Table 5.

**Table 5.** An example of heuristic action c in educational research.

| Conversation Excerpt | Descriptive Label (Singular Concrete Essence) |
| --- | --- |
| 'When I am ill, mum gives me a medicine that I always like'. | Medicating people |
| 'I care for my cats as well. I hold them in my arms; I protect them'. | Protecting others |

The descriptive labels were finally quantified (see the column N. in the Tables 6–8) to understand the children's main avenues of thought.

**Table 6.** An example of heuristic actions d and e in educational research.

| Descriptive Label (Singular Concrete Essence) | N. | Category Label (First-Level Extended Essence) |
|---|---|---|
| Medicating people | 26 | Healing injuries |
| Recalling a condition of discomfort | 12 | |
| Allowing the other to do what they like | 5 | Making others feel well |
| Promoting happiness | 4 | |
| Offering reassurance | 2 | |

**Table 7.** Coding system of results for heuristic action f in educational research.

| Example of Data | Descriptive Label (Singular Concrete Essence) | N. | Category Label (First-Level Extended Essence) |
|---|---|---|---|
| 'Someone got injured . . . to care means that he went to the medical doctor'. | Medicating people | 26 | Healing injuries |
| 'When I [sprained] my finger and my foot'. | Recalling a condition of discomfort | 12 | |
| 'Care is . . . when I feed the fishes with Mariella'. | Feeding others | 18 | Preserving life |
| 'I also care for my cats. I hold them in my arms; I shelter them'. | Protecting others | 5 | |
| 'Because my dad always lets me play [with the Xbox], and then [if I waste the batteries] he lends them to me'. | Allowing the other to do what they like | 5 | Making others feel well |
| 'He has made him happy'. | Promoting happiness | 4 | |
| 'It is when I go to bed because my mum holds my hand until I get asleep'. | Offering reassurance | 2 | |
| 'When there is a child who is a kid and thinks that toys get broken . . . and . . . they . . . they . . . he must take care with them and when they break . . . he repairs them'. | Respecting others | 14 | Practicing solicitude |
| 'To care is when I care for my puppy, Bianca. I take her out with my dad or with uncle Vale'. | Paying attention to the needs of the other | 14 | |
| 'I care for my dad. I give him many kisses'. | Making gestures of affection | 10 | Manifesting affection |
| 'Care means "I love you"'. | Loving others | 2 | |

    d.    *Build classes of similar data and*
    e.    *Formulate the first level of the extended essences*

Starting from a list of descriptive labels that expressed the singular concrete essences of the lived experiences of care reported in the children's answers, classes were elaborated. Each class, or category, groups descriptive labels referring to similar meanings of 'care' and expresses the extended concrete essence shared by them. Each extended essence is expressed with a concept or category label. An example of this clustering and conceptual labeling action is presented in Table 6.

**Table 8.** Final coding system of educational research, integrated in accordance with heuristic action i.

| Descriptive Labels (Singular Concrete Essences) | N. | Category Label (First-Level Extended Essence) |
|---|---|---|
| Feeding others:<br>- Feed others;<br>- Provide water. | 18 | Preserve life |
| Protecting others:<br>- Provide shelter;<br>- Defend others;<br>- Alert others to danger. | 5 | |
| Respect others:<br>- Treat toys well;<br>- Repair toys;<br>- Treat animals well;<br>- Care for plants. | 14 | Show solicitude |
| Pay attention to the needs of the other:<br>- Help those in trouble;<br>- Play together with one's younger sister;<br>- Be very patient with one's younger brother;<br>- Walk the puppy;<br>- Pay attention when choosing food to feed animals;<br>- To care is to feel the heart of the other. | 14 | |

f.  *Build a hierarchy of essences*

In this study, no second-level extended essences were found: indeed, the goal of achieving the most general level of the concrete essence of care disclosed by the children's answers was reached through the elaboration of the categories. In light of this, the final coding system, which shows the hierarchy of essences emerged in this research, includes only descriptive and category labels; it does not include any macrocategory labels because second-level clusters, i.e., classes of categories, were not elaborated (Table 7).

g.  *Recover and describe the absolutely unique data*

Some of the children's ideas about care were outliers; because of the richness and deepness of their contents, to codify these data would inevitably reduce their expressive potential. Since these outlying data express an indivisible semantic synthesis, they can be considered complex thoughts that were not included in the coding system. Instead, they were classified in the research report as unique data, i.e., data that revealed essential content not shared by anyone else.

Below, we present examples of such outlying data, briefly focusing on the reasons for their uniqueness:

'It is me and Sofi who play. I with the toy car, and she with the doll. If she does not scratch me, we feel very good'.

In this answer, care is described by the child as a way of feeling good that takes place at the moment when one does not get injured. The complexity of the idea expressed by this datum is created by the combination of the concept of care with that of feeling well, which occurs thanks to forms of protection—feeling preserved and safe inside a relationship.

'If you find some sees, do not go there because if you find some thorns and crocodiles and they eat you, and otherwise I come and care for you very much, and I love you so much'.

In this answer, the child expresses, in a complex way, the dimension of having the other at heart. The first described action is a warning about danger that tries to preserve the other

from the injuries that they could get ('do not go there'); the child then continues with a declaration of a therapeutic action in case of pain ('and otherwise I will come and care for you very much'); finally, he ends with a concise declaration of what motivates this sort of actions: affection ('I love you so much'). What makes this datum a unicum, and thus not codifiable into the elaborated coding system, is the idea that care consists in a form of loving the other that is aimed to preserve them or heal their injuries.

h. *Elaborate the descriptive theory*

Finally, the application of the EPM in this research allowed the elaboration of a descriptive theory about how care is conceived and lived by the involved kindergarten's children. The coding system which emerged from the data analysis, addressed the writing of a text which described the phenomenon of care as healing injuries, preserving life, making others feel well, practicing solicitude and manifesting affection. In writing down the theory, these categories, which express the extended concrete essences found through data analysis, have been deepened with reference to the labels, i.e., the individual concrete essences, clustered in them, and with reference to the collected data, i.e., the original ideas of children, in order to make the connection evident between the level of conceptualization and the level of experiential evidence. Furthermore, the absolutely unique data were presented and commented, as highlighted above (see above, heuristic action g).

i. *Implement the principle of recursiveness*

The continuous return to the data collected and the descriptive and conceptual labels formed, designed to verify the researchers' capacity to faithfully express in the coding system the singular and extended concrete essences of care individuated in the children's answers, suggested for some of the elaborated descriptive labels the possibility of extracting a further level of singular essences more concrete than the ones expressed through the first labeling process (see heuristic action c).

Below, we present the pieces of the coding system, which have been integrated with the specifications of the descriptive labels expressing this further level of concrete singular essences found by applying the principle of recursiveness (Table 8).

## 6. Discussion

This research has highlighted that the application of the EPM is not simple because the goal of defining the essence of the concrete in a rigorous way is difficult to achieve. For this aim, a series of research procedures and analysis techniques support researchers in giving form to a phenomenologically based descriptive theory. However, the heuristic actions and the methodological principles identified and exemplified above, should be conceived as flexible "guidelines" for the researcher. Differently from the tendency to codify procedures to conduct phenomenological empirical research as an answer to the request for strict methodological scientific principles, as in Moustakas (1994), (Giorgi 1997; Giorgi and Giorgi 2003) and Ashworth's (Ashworth 2003; Ashworth et al. 2003) proposals, we agree with the position of Van Manen (1990), Bengtsson (2013b), Dahlberg (Dahlberg 2006; Dahlberg and Dahlberg 2019) and Zahavi and Martiny (2019) who invoke for a less systematic and more creative approach to the method. In fact, if the motto of the Husserlian approach is to 'go back to the "things themselves"' (Husserl 2001, p. 168) this means that the researcher has to follow things as they appear and not a particular method, conceived as a series of strictly codified research procedures and analysis techniques.

In order to found an empirical phenomenological theory, it is important to deeply know the phenomenological philosophy by being familiar with the original sources of phenomenology. This is because as Zahavi (2019) states, 'phenomenologically informed qualitative research has different aims than phenomenological philosophy, but it is questionable whether the former can qualify as phenomenological if it either ignores or misinterprets the latter' (p. 900). Therefore, as with some of the most influential scholars quoted in the introduction of this article, in order to provide a rigorous foundation to our phenomenological empirical method we refer directly to the original Husserlian production, and not to

secondary sources. However, we recognise that the eidetic concepts, which found philosophical phenomenology, are not simply adaptable to the empirical field. Indeed, we refer to some concepts, i.e., the ones of the essence of concrete and the subsequent adumbrations, highlighted in the first of Husserl's production that the author himself suggests for the knowledge of concrete lived experience. With regards to other qualitative research approaches, even if not strictly phenomenologically grounded, we can find some similarities in the procedures suggested to analyse, conceptualize and cluster data. This happens, for example, in comparison with the thematic (Clarke and Braun 2021; Terry et al. 2017) and grounded theory (Glaser and Strauss 1967; Strauss and Corbin 1998; Charmaz 2014) analysis approaches. However, the specificity of the EPM lies in the concern of rooting the research practices in a rigorous theoretical framework, which includes ontological, gnoseological and epistemological assumptions.

Giving a critical glance to the work requested to a researcher following the EPM, we can highlight some difficulties that can hardly be overcome. In the processes of labeling, the greatest difficulty consists in finding the words to faithfully express the individual and extended qualities of the collected lived experiences; indeed, the concrete flow of an experience does not lend itself to be grasped in an utterance with well-defined contours. The purpose is to 'bring to the normal distance, to complete clearness, what at any time floats before us shifting and unclear' (Husserl 2012, p. 131). The labels must be clearly distinguished from one another in order to exclude possible confusion in the interpretations of the final findings. The formulation of a concrete essence can be considered adequate when it allows precise reference to the collected lived experiences. This implies that researchers should be guided by the principle of faithfulness, which requires them to patiently ensure the adherence of the words to the data. The epistemic effort to find faithful words, i.e., the words that express the concrete essence of the investigated phenomenon in the most appropriate manner with regard to the collected data, is also an ethical effort, because it is an action of respect towards the research participants. In order to fulfil the task of achieving the highest possible faithfulness to the data, it is important to carry out the epistemic act of 'epochè' as well as to discuss the labels formulated within the research group so that each individual perspective can be enriched by the perspective of the others.

The outcome of a research project carried out according to the EPM is the description of the investigated phenomenon using a concretely rich concept that includes all the qualities found through the analysis of experience. The achievement of this outcome cannot be conditioned by a predefined timeline because the principle of keeping the thought connected to reality in its concrete occurrence leads the researcher to work on a continuous and repeated definition of the descriptive theory. Only general truths, and thus eidetic essences, which impose themselves as evident, have a definitive value, while the statements about the world of experience remain open to the possibility of continuous reformulation.

The insuperable limitation of the EPM is its focus on the particular qualities of the investigated phenomena, which makes it difficult, if not impossible, to achieve knowledge of general value. The truth that can be reached is a local and situated truth, but if it is achieved through a rigorous method, it can nonetheless meet the need to deeply understand the human lived experience, which cannot be enlightened by general and abstract knowledge. Even if the EPM does not allow us to reach general knowledge—indeed, this is the purpose of an eidetic science and not an empirical one—rigorous phenomenological empirical research can yield concepts able to describe not only the investigated lived experiences but also crucial aspects of new lived experiences of the same phenomenon that have not yet been subjected to examination.

## 7. Conclusions

Starting from the consideration that the application of phenomenology, which was founded as a philosophical method, to the empirical field needs to be furtherly thought out, this study circles back to some of Husserl's original concepts in order to explore the possibility of rigorously founding an empirical phenomenology. This goal has been fulfilled thanks

to a discussion of some fundamental concepts of phenomenological philosophy, i.e., the concepts of essence and intuition. The EPM that has been consequently grounded, focuses on the Husserlian concepts of concrete essence and subsequent adumbrations—respectively, the object and the method of an empirical phenomenological investigation. Indeed, the human and social sciences, as sciences of facts, do not search for the eidetic essences of phenomena, i.e., their general and necessary predicates, but are interested in their concrete, singular, contextual and accidental qualities. Therefore, intuition, conceived by Husserl as the immediate apprehension of a phenomenon's eidetic essence, cannot be the fundamental cognitive act of an empirical phenomenological investigation; instead, the concrete essence of a phenomenon can be reached gradually, by subsequent adumbrations, which put into action a continuous attention to the phenomenon so it can be faithfully understood.

Since the theoretical foundation of an empirical method is not enough to make possible a discussion of its effectiveness, this article discussed the application of the EPM in two different fields, education and healthcare. The different steps of this application were carefully presented to make evident how the previously described heuristic actions and epistemic principles take form in a concrete research process. The application of the EPM to the presented studies showed its effectiveness with respect to the purpose of carrying out scientific research, which can be useful to the improvement of nursing and educational practices. Indeed, the conceptualisation of good care in nursing outlined in the first example can effectively contribute to the development of good caring practices in healthcare structures. On the other hand, the findings about the richness of children's ethical thinking that emerged from the second inquiry demonstrate the educational effectiveness of carrying out Socratic conversations with children. In conclusion, the EPM can be conceived as a rigorous method in the empirical research field as well as an effective way to carry out research with useful practical implications.

**Author Contributions:** Conceptualization of the EPM, L.M.; application of the EPM in the healthcare field, L.M. and R.B.; application of the EPM in the educational field, L.M., M.U. and F.V.; writing—original draft, L.M., R.B., M.U. and F.V.; writing—review and editing, R.B., M.U. and F.V.; supervision, L.M. All authors have read and agreed to the published version of the manuscript.

**Funding:** The research presented in the Section 5.2 as an example of application of the EPM in the educational field was funded by the Italian Ministry of Education, University and Research.

**Institutional Review Board Statement:** Not applicable.

**Informed Consent Statement:** Informed consent was obtained from all subjects involved in the studies presented in the Sections 5.1 and 5.2 as examples of application of the EPM in the healthcare and educational fields.

**Conflicts of Interest:** The authors declare no conflict of interest.

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
