# Peer review of "The Empirical Phenomenological Method: Theoretical Foundation and Research Applications"

_socsci, doi:10.3390/socsci12070413_

Round 1

Reviewer 1 Report

Review: The empirical phenomenological method: Theoretical foundation and research applications

This article about empirical phenomenological method has great ambitions, however the text does not meet all these expectations regarding theoretical foundation and development of a dense new method. In the text below I have tried to outline some parts which I find valuable to develop further.  

The introduction gives a lot of references to authors who have developed and discussed some kind of phenomenological method. However, the examples are very old and nothing is mentioned from the last two decades. The references need to be updated and complemented with newer empirical methods and developed methodology. I also suggest that there is not enough distinction between phenomenology as a philosophical method and phenomenology as an empirical method. You also write that the empirical sciences are sciences of facts, but I do not completely agree; are phenomenological empirical research not studies of the lived world, i.e. things as phenomena, and not objective facts. The experience of something is most often present in this kind of empirical research. But perhaps I have misunderstood your claim.

When it comes to methodology and the use of theory from phenomenological philosophers, actually, there are scholars who have developed methodology grounded in philosophical theory; to name a few Jan Bengtsson (2013), Karin Dahlberg, van Manen and Gunnar Karlsson (EPP method), as well as Giorgi who is mentioned in the text. They have all contributed to the field of empirical phenomenology, but with different focus within the phenomenological tradition. Their research and development of epistemology are also of great relevance and relates also to the fields of education and nursing. Another scholar that has contributed widely to the discussion of phenomenology and its use in qualitative research is Dan Zahavi, for example in an article within Qualitative Health Research 2018. His discussion about phenomenological philosophical method in comparison to phenomenological empirical methods would be of great importance and enhance the discussion in the introductory paragraphs. With these references, the argument that the phenomenology has not been adequately thought on will fail. Besides, the argumentation in line with Cohen and Omery, is from a text almost 30 years ago.

It is also a bit odd that Heidegger is referred while he did not agree to Husserl’s transcendental philosophy, which is the focus in the manuscript.

I agree with you when you write that there must be other methods for empirical research and that philosophical methods should not be equivalent with empirical methods. But, how could it be that outlining a theory has to identify similarities and differences between eidetic phenomenology and empirical phenomenology; should not the case instead be to develop theory as a starting-point for empirical research? You also write about the object, but on other places in the text, and in accordance with phenomenology, you emphasize the concept of phenomenon as basic. You also talk about the experiences of the mind, but if you mean concrete experiences, mustn’t there also be an individual, a person with a body? There are quite few explicit references to Husserl in the theory section, and it could be informative with some lines regarding which period of Husserl that you rely on in explicating theory. As an overall comment it is not so easy to understand the theoretical part of the manuscript. This part of the manuscript could be clearer in defining the theory for the empirical phenomenology, so it will be easier to understand, also for those who are not so acquainted with phenomenological theory in its eidetic forms and reductions. In section 2.1. you write about methodology, but it is not so easy to follow what you mean by knowledge related to singular concrete essences in relation to extended essence of the concrete.

Sometimes it is difficult to understand if you relate to the philosophical part, or the more empirical aspect of something. The paper should be more understandable if you could do that clearer. It should also perhaps be of importance if you somewhere in the text could give a short overview of Husserl’s various phases in developing the theory of phenomenology, as they differ from each other. When writing about the epoché, eidetic intuition and other central concepts it should have been valuable if you had related to in which other empirical phenomenological methods or methodologies these concepts are also used. It should also be noticed that the section of theory also includes methodology, but perhaps you relate to this as epistemology (epistemological theory). As a conclusion, it should be clearer how the outlined theory guides the developed EPM.

In section 3, it should be better described what are the three components in the table, and what it expresses. It is quite difficult to understand what is described in the table, without referring to any data. You write about how your model has taken in the previous tradition within the phenomenological tradition, but this needs to be more explicit with examples and references, and not just to refer to all the researchers together. It also needs to be clearer how grounded theory, which is not a phenomenological tradition, has informed your developed method. This is essential when dealing with development of methodology.

How is Figure 1 related to Table 1? It is not quite obvious, could that be explained in more detail? Further, how are the categories and macro-categories developed? How relate these to singular essences?

When it comes to the EPM in healthcare research this section could be complemented by methodological or methods references. Some concepts could also be elaborated on, as for example eidetic interviews. I strongly agree in the choose of narratives, but shouldn’t it be better to argue for narratives according to phenomenology and not to other research and scholars. Note that also here the references are only older texts.

The argumentation regarding the number and choosing of participants are not primarily phenomenological, as how to reach the chosen phenomenon and its variation. This could be elaborated.

As the procedure is described, it seems that there are great similarities with some kind of thematic analysis, or Giorgi’s recommended analysis; have you reflected on this, or do you see your method as completely new?

My suggestion is to condense table 4, it is quite extensive and perhaps not all of the examples are needed in order to describe this phase of the analysis.

When the educational example is presented, it could have been mentioned that the presentation follows the same headings as the earlier example of care.

One question about the material left for the children’s interviews; could it be so that the left quotations actually did not fit in the developed system of different labels?

It is a little bit surprising that the discussion does not integrate an examining of this intended developed EPM in relation to other well-defined methods, such as for example grounded theory and thematic analysis https://www.thematicanalysis.net/doing-reflexive-ta/ Also, other phenomenological approaches could be reflected upon. Interpretation is also mentioned, but earlier in the text, description has been the guiding star.

While the EPM is said to be grounded in the philosophy of Husserl, this could have been reflected more upon in the discussion or in the conclusions, for example how it has been possible to catch the essence of the chosen phenomena.  

As a summary, I will recommend some further work with the manuscript; especially to tighten up the theoretical part so it more clearly relates to the development of method, but also nuance the new method developed and describe what is new and/or unique, in relation to other earlier developed approaches in the field.

Author Response

Response to Reviewer 1 Comments 

Point 1.The introduction gives a lot of references to authors who have developed and discussed some kind of phenomenological method. However, the examples are very old and nothing is mentioned from the last two decades. The references need to be updated and complemented with newer empirical methods and developed methodology.  

Response 1. The introduction has been updated and complemented with new and more recent references. 

Point 2. I also suggest that there is not enough distinction between phenomenology as a philosophical method and phenomenology as an empirical method.  

Response 2. Within the text we have tried to better explain this difference (in particular see Introduction, paragraphs 2 and 3) 

Point 3. You also write that the empirical sciences are sciences of facts, but I do not completely agree; are phenomenological empirical research not studies of the lived world, i.e. things as phenomena, and not objective facts. The experience of something is most often present in this kind of empirical research. But perhaps I have misunderstood your claim. 

Response 3. In the Introduction we used the notion “science of ‘fact’” referring to Husserl’s idea that “science of fact and science of experience are equivalent concepts” (Ideas, par. 2-7). In the reviewed text we have explicitly given reference to Husserl. 

Point 4. When it comes to methodology and the use of theory from phenomenological philosophers, actually, there are scholars who have developed methodology grounded in philosophical theory; to name a few Jan Bengtsson (2013), Karin Dahlberg, van Manen and Gunnar Karlsson (EPP method), as well as Giorgi who is mentioned in the text. They have all contributed to the field of empirical phenomenology, but with different focus within the phenomenological tradition. Their research and development of epistemology are also of great relevance and relates also to the fields of education and nursing. Another scholar that has contributed widely to the discussion of phenomenology and its use in qualitative research is Dan Zahavi, for example in an article within Qualitative Health Research 2018. His discussion about phenomenological philosophical method in comparison to phenomenological empirical methods would be of great importance and enhance the discussion in the introductory paragraphs. With these references, the argument that the phenomenology has not been adequately thought on will fail. Besides, the argumentation in line with Cohen and Omery, is from a text almost 30 years ago.  

Response 4. We have integrated the article with a new paragraph wich presents an overiew of some of the main empirical phenomenolgical approaches. In particular we gave reference to Bengtsson, Dahlberg, Karlsson, Giorgi, Van Manen, Moustakas and Zahavi. Furthermore ,we have more clearly presented the debate about the phenomenological philosophical method in comparison to phenomenological empirical methods, by giving reference to Crotty, Giorgi and Zahavi. 

Point 5. It is also a bit odd that Heidegger is referred while he did not agree to Husserl’s transcendental philosophy, which is the focus in the manuscript.  

Response 5. Even if our theory of empirical phenomenology is primarily grounded in Husserl’s philosophy with particular reference to some concepts highlighted in the Ideas, we recognize the influence of Heidegger’s philosophy in other empirical phenomenological approaches (for example , Heideggerian Hermeneutics School) 

Point 6. I agree with you when you write that there must be other methods for empirical research and that philosophical methods should not be equivalent with empirical methods. But, how could it be that outlining a theory has to identify similarities and differences between eidetic phenomenology and empirical phenomenology; should not the case instead be to develop theory as a starting-point for empirical research?  

Response 6. In order to develop a theory as a starting-point for empirical phenomenological research, we considered important to highlight the difference between eidetic phenomenology and empirical phenomenology. This clarification seemed necessary to us consequently to our statement that phenomenology has originally been developed by Husserl as a method for eidetic science and not for empirical science. However, we have rephrased this part for a clearer explanation. 

Point 7. You also write about the object, but on other places in the text, and in accordance with phenomenology, you emphasize the concept of phenomenon as basic.  

Response 7. In the previous statement “the object is the same - namely the experience of the mind”, we intended to say that phenomenology is focused on the lived experiences of the mind; we used the term “object” as synonymous of “focus”. Now we have rephrased in a more appropriate way. 

Point 8. You also talk about the experiences of the mind, but if you mean concrete experiences, mustn’t there also be an individual, a person with a body? 

Response 8. We have inserted in the paragraph 3 a reference to Merleau-Ponty and the consideration of the bodily dimension.  

Point 9. There are quite few explicit references to Husserl in the theory section, and it could be informative with some lines regarding which period of Husserl that you rely on in explicating theory.  

Response 9. We have specified that we refer to the first Husserlian production, and in particular to Ideas (see Introduction). 

Point 10. As an overall comment it is not so easy to understand the theoretical part of the manuscript. This part of the manuscript could be clearer in defining the theory for the empirical phenomenology, so it will be easier to understand, also for those who are not so acquainted with phenomenological theory in its eidetic forms and reductions.  

Response 10. The addition of the new paragraph 2 allows us to better present and explain the most important phenomenological concepts and principles.  

Point 11. In section 2.1. you write about methodology, but it is not so easy to follow what you mean by knowledge related to singular concrete essences in relation to extended essence of the concrete.  Sometimes it is difficult to understand if you relate to the philosophical part, or the more empirical aspect of something. The paper should be more understandable if you could do that clearer.  

Response 11. We have improved the empirical part of the article to better clarify and exemplify the concepts and principles explained in the theoretical one.  

Point 12. It should also perhaps be of importance if you somewhere in the text could give a short overview of Husserl’s various phases in developing the theory of phenomenology, as they differ from each other. When writing about the epoché, eidetic intuition and other central concepts it should have been valuable if you had related to in which other empirical phenomenological methods or methodologies these concepts are also used.  

Response 12. We specified that we refer to the first Husserlian phase (Ideas), and we presented our reflection in dialogue with other phenomenological approaches (see par. 2 and Discussion) 

Point 13. It should also be noticed that the section of theory also includes methodology, but perhaps you relate to this as epistemology (epistemological theory). As a conclusion, it should be clearer how the outlined theory guides the developed EPM.  

Response 13. If with “epistemology” you mean a set of principles which allows a rigorous foundation of a research, then we can intend our theory as an epistemology.  

Point 14. In section 3, it should be better described what are the three components in the table, and what it expresses. It is quite difficult to understand what is described in the table, without referring to any data.  

Response 14. We improved the table.  

Point 15. You write about how your model has taken in the previous tradition within the phenomenological tradition, but this needs to be more explicit with examples and references, and not just to refer to all the researchers together.  

Response 15. We improved our references. 

Point 16. It also needs to be clearer how grounded theory, which is not a phenomenological tradition, has informed your developed method. This is essential when dealing with development of methodology.  

Response 16. We eliminated the reference to the grounded theory in this paragraph, and we added a comparison with the grounded theory in the Discussion.  

Point 17. How is Figure 1 related to Table 1? It is not quite obvious, could that be explained in more detail? Further, how are the categories and macro-categories developed? How relate these to singular essences?  

Response 17. We improved the link between figure 1, table 1 and the concepts explained in the text. 

Point 18. When it comes to the EPM in healthcare research this section could be complemented by methodological or methods references. Some concepts could also be elaborated on, as for example eidetic interviews.  

Response 18. We integrated the text consistently with an improvement of the table of the heuristic actions, but we would like to underline that this part is intended as an exemplification of the method presented above. 

Point 19. I strongly agree in the choose of narratives, but shouldn’t it be better to argue for narratives according to phenomenology and not to other research and scholars. Note that also here the references are only older texts.  

Response 19. We implemented this part with reference to phenomenologists. 

Point 20. The argumentation regarding the number and choosing of participants are not primarily phenomenological, as how to reach the chosen phenomenon and its variation. This could be elaborated.  

Response 20. We referred to the concept of “purposeful sampling” (Merriam, 2002), as a concept shared by other phenomenological approaches (participants who have a “practical understanding” of the phenomenon: see Bengtsson). 

     

Point 21. As the procedure is described, it seems that there are great similarities with some kind of thematic analysis, or Giorgi’s recommended analysis; have you reflected on this, or do you see your method as completely new?  

Response 21. We confirm that the EPM has similarities to other phenomenological approaches. We added some references and stressed the specificities of our method. 

Point 22. My suggestion is to condense table 4, it is quite extensive and perhaps not all of the examples are needed in order to describe this phase of the analysis.  

Response 22. In order to show the faithfulness of our analytical procedure  and its products to the original data, we consider necessary to present an example for every label expressed. However, in order to make table 4 more readable, we can ask to the journal editor to improve the layout, for example by enlarging the first column.  

Point 23. When the educational example is presented, it could have been mentioned that the presentation follows the same headings as the earlier example of care. 

Response 23. We explained the parallelism between the two examples. 

Point 24. One question about the material left for the children’s interviews; could it be so that the left quotations actually did not fit in the developed system of different labels? 

Response 24. The protruding data are left out of the coding system and presented in the paragraph corresponding to the heuristic action (g).  

Point 25. It is a little bit surprising that the discussion does not integrate an examining of this intended developed EPM in relation to other well-defined methods, such as for example grounded theory and thematic analysis https://www.thematicanalysis.net/doing-reflexive-ta/ Also, other phenomenological approaches could be reflected upon. Interpretation is also mentioned, but earlier in the text, description has been the guiding star.  

Response 25: We integrated the discussion paragraph deepening comparison with other qualitative approaches. 

Point 26. While the EPM is said to be grounded in the philosophy of Husserl, this could have been reflected more upon in the discussion or in the conclusions, for example how it has been possible to catch the essence of the chosen phenomena.   

Response 26. Our text has been improved to better clarify how EPM can guide the researcher in discovering the concrete essence of the investigated phenomenon. Moreover, in the conclusion paragraph we stressed the difficulties for an empirical researcher in catching the essence. 

Point 27. As a summary, I will recommend some further work with the manuscript; especially to tighten up the theoretical part so it more clearly relates to the development of method, but also nuance the new method developed and describe what is new and/or unique, in relation to other earlier developed approaches in the field.  

Response 27. As a summary, we hardly worked at the improvement of the article, trying to answer to every critical points suggested. We sincerely thank the reviewer for the precise, punctual and very competent critics and suggestions. Thanks to him/her, we lived the revision work as an opportunity for creating a “community of research”, even if at distance and in a “blinded way”. 

Reviewer 2 Report

This is a very interesting, original and carefully crafted paper. I particularly appreciate the clear theoretical positioning (Husserl), which then leads into the two examples of qualitative studies. However, it might be helpful to state even more clearly which aspects of Husserl's work the authors refer to: Husserl's work/ideas transitioned throughout his life; thus, it would be helpful to know which works, stages of his work, concepts, etc. you are referring to. In addition, you might want to position your take on phenomenological research (which is based on Husserl) within the existing phenomenological research approaches (e.g. Max van Manen, etc.), i.e. existing approaches that might refer to a different phenomenological theories, e.g. Merleau-Ponty, Heidegger, etc. Otherwise, I congratulate you on this paper and I will gladly use it in my courses on phenomenological research.

Author Response

Response to Reviewer 2 Comments

Point 1. This is a very interesting, original and carefully crafted paper. I particularly appreciate the clear theoretical positioning (Husserl), which then leads into the two examples of qualitative studies. However, it might be helpful to state even more clearly which aspects of Husserl's work the authors refer to: Husserl's work/ideas transitioned throughout his life; thus, it would be helpful to know which works, stages of his work, concepts, etc. you are referring to.

Response 1. We improved the article by specifying the sources in Husserl’s production. In particular we refer to Ideas.

Point 2. In addition, you might want to position your take on phenomenological research (which is based on Husserl) within the existing phenomenological research approaches (e.g. Max van Manen, etc.), i.e. existing approaches that might refer to a different phenomenological theories, e.g. Merleau-Ponty, Heidegger, etc. Otherwise, I congratulate you on this paper and I will gladly use it in my courses on phenomenological research.

Response 2. We have added a new paragraph which presents an overiew of some of the main empirical phenomenolgical approaches. In particular we gave reference to Bengtsson, Dahlberg, Karlsson, Giorgi, Van Manen, Moustakas and Zahavi. Furthermore we have more clearly presented the debate about the phenomenological philosophical method in comparison to the phenomenological empirical methods, by giving reference to Crotty, Giorgi and Zahavi. 

Reference to other phenomenological philosophers have been added.

Moreover, we have also included in the paragraph of discussion some new references to other phenomenological approaches in order to better clarify the specificities of the EPM.

Round 2

Reviewer 1 Report

Thank you for the revised manuscript. I have read through all the responses to the earlier comments and find that all of the them are clearly addressed. The manuscript is now much clearer and will make an important contribution to the field of empirical phenomenological research.

When I read through the document, I found some details in the text which perhaps also could be taken into consideration and addressed:

Line 73. Fleming is missed in the references

Line 83-88. This sentence is difficult to understand. It is quite long and includes quotes to several researchers. Could it be split into two sentences, as a suggestion, and perhaps also made clearer in its’ argumentation.

Line 109-113. Is Karlsson still referred here?

Line 206-208. You write that what is problematic is that most existing empirical phenomenological approaches have been drawn on concepts thought up by Husserl for phenomenology as an eidetic research method. However, in the text by Bengtsson (2013 With the lifeworld as ground. A research approach for empirical research in education) the lifeworld approach is not described as an eidetic research method.

Line 311-315. I find this sentence hard to understand, could it be made clearer. Which other research empirical philosophies do you relate to?

Line 380. Do you have any reference to eidetic interviews?

Line 444. In the reflective supervision, is there any other researcher present in the process and/or analysis? Or, how do you work with this principle in the research team?

Line 445. Arendt is not present in the references.

Line 577 “were classified in the research report” Are these results presented originally in any other publication or presentation?

Author Response

Point 1: Line 73. Fleming is missed in the references

Answer 1: the reference has been added.

Point 2: Line 83-88. This sentence is difficult to understand. It is quite long and includes quotes to several researchers. Could it be split into two sentences, as a suggestion, and perhaps also made clearer in its’ argumentation.

Answer 2: the sentence has been re-formulated.

Point 3: Line 109-113. Is Karlsson still referred here?

Answer 3: the reference has been clarified.

Point 4: Line 206-208. You write that what is problematic is that most existing empirical phenomenological approaches have been drawn on concepts thought up by Husserl for phenomenology as an eidetic research method. However, in the text by Bengtsson (2013 With the lifeworld as ground. A research approach for empirical research in education) the lifeworld approach is not described as an eidetic research method.

Answer 4: We agree with the reviewer’s comment where he specifies that, according to Bengsston, the concept of lifeworld is applied to empirical research. What we intended to stress, and it is confirmed by the reviewer, is that this concept, too, comes from Husserl’s eidetic position and it is applied to the empirical field. Thank you for this suggestion.

Point 5: Line 311-315. I find this sentence hard to understand, could it be made clearer. Which other research empirical philosophies do you relate to?

Answer 5: we added some examples (with references) in order to make the sentence clearer.

Point 6: Line 380. Do you have any reference to eidetic interviews?

Answer 6: The distinction between eidetic and narrative interviews is proposed by the authors, and it consists in the different questions asked to the participants: the former is a question which requires to describe the essence of the investigated phenomenon , the second is a question which requires to narrate the experience of the phenomenon.

Point 7: Line 444. In the reflective supervision, is there any other researcher present in the process and/or analysis? Or, how do you work with this principle in the research team?

Answer 7: we added some lines in order to narrate the role of reflective supervision.

Point 8: Line 445. Arendt is not present in the references.

Answer 8: the reference has been added.

Point 9: Line 577 “were classified in the research report” Are these results presented originally in any other publication or presentation?

Answer 9: Yes, the complete report of the two researches presented are published, as correctly referred in the first lines of paragraph n. 5. The purpose of the present paper is different, because it is aimed at exemplifying the EPM described in the first part.

In conclusion, we would like to thank the reviewer for these further precious suggestions and the care he acted towards our paper!